# Culturally Relevant STEM (CReST): An Integrated Support Curriculum for High School Chemistry and World History

**James K. Ferri * and Rachel Sparks White**

Department of Chemical and Life Science Engineering, Virginia Commonwealth University, Richmond, VA 23219, USA; whiter7@vcu.edu
* Correspondence: jkferri@vcu.edu

**Abstract:** Convergence education, driven by compelling or complex socio-scientific problems, is an approach to bring cultural relevance into secondary STEM education. National trends show the need to increase the STEM workforce by leveraging educational research and innovative practices within the secondary level to increase student interest prior to graduating high school. We introduced CReST (Culturally Relevant STEM) in a US high school pilot study. Student participants included 276 Chemistry students and 19 World History I students. The study also engaged four (4) high school teachers in chemistry and social studies with the challenge of cultural heritage conservation through the lens of the (physicochemical) life cycle of mural paintings in Europe. Teachers were provided with (1) professional development; (2) a digital curriculum; and (3) modular kits for hands-on learning. The research focused qualitatively on the experiences from the teacher and students as well as quantitatively to assess whether there was an increase in student academic performance. We found a statistically significant gain with respect to Chemistry (4.0%) and World History (13.4%) content. Students and teachers responded with overwhelming positivity in individual and focus-group interviews. This amplifies the further need of convergent educational approaches in high school STEM education to enhance engagement and increase student learning.

**Keywords:** STEM education; transdisciplinary learning; convergence education; experiential learning; conservation science; culturally relevant pedagogy

## 1. Introduction

Societal demands on the global workforce display an increasing need for individuals to master science, technology, engineering, and mathematics (STEM) [1]. According to the U.S. National Science Board indicators in 2021, the U.S. STEM workforce includes 16 million workers with an education at the bachelor's degree level and nearly 20 million workers who do not have a bachelor's degree in the skilled technical workforce [2]. For demographic groups, there continues to be an uneven representation in the STEM workforce. This trend seems to be most pronounced for girls and particular non-white ethnic minorities [3–5]. Participation in advanced STEM coursework in secondary and higher education, and improvements in facilitating and sustaining STEM interest and participation, particularly amongst those in poor, under-resourced communities, have been at best modest [6]. These are indicators that opportunities to increase the STEM workforce with domestic talent [7] should be a focus on STEM educational research and innovative practices within the secondary level to increase student interest prior to graduating high school.

Convergence education is driven by compelling or complex socio-scientific problems or topics, where learners apply knowledge and skills using a blended approach across multiple disciplines (i.e., transdisciplinary) to create and innovate new solutions [8]. Widya et al. discuss the importance of making connections between what is taught in the STEM classroom and what is occurring in daily life to improve student learning by [9]. Science communities are striving to revise the traditional process of learning where students follow

step-by-step procedures to perform a given task to reach a conceptual understanding of the content [10]. The transition from step-by-step labs to a more problem-based and inquiry model represents more of an integrated STEM approach to learning. The integration of engineering design practices into science education allows students to solve relevant real-world challenges that our society faces. The Next Generation Science Standards (NGSS) is committed to leveling engineering design with scientific inquiry within elementary and secondary education [11]. To address these needs, we posit a modular, experiential transdisciplinary STEM support curriculum that addresses a global and cultural thread through cultural heritage conservation. Culturally relevant STEM (CReST) is a transdisciplinary learning program that leverages artifacts of cultural heritage as boundary-crossing objects to enable students and teachers to connect key concepts of science, technology, engineering, and mathematics (STEM) to their personal and collective background and experience through convergence education. CReST aims to increase the diversity, equity, and interest of student learners through culturally relevant teaching and structured, hands-on, and transdisciplinary learning.

Through these experiences, students are exposed to artifacts that connect traditionally siloed educational experiences in science, social studies, and engineering in multiple days of instruction. CReST features (1) a convergence educational framework; (2) partnerships with highly visible organizations and institutions providing professional development workshops for participating teachers and career exposure to students; and (3) a strategic global context, focusing on culturally relevant pedagogy, transdisciplinary learning, real-world problem solving, and communities of practice.

The aim of this research was to examine contributory factors to teaching and learning Chemistry and World History content following a six-lesson day intervention that featured the lifecycle of a form of mural art, the fresco, as a case study in materials science, inorganic chemistry, environmental chemistry, and cultural heritage conservation. The purpose of this pilot study was to develop, implement, and assess a new support curriculum, CReST, that leverages cultural heritage in Chemistry and World History I high school education using the fresco lifecycle as an experiential case study. With the fresco artifact serving as the boundary-crossing object, the convergence of Chemistry and World History was an organic pairing for this study. The goals of this study were to understand the learning experiences of the teacher and student participation in the CReST support curriculum through a professional development (PD) workshop and classroom intervention. An assessment of high school student academic performance in Chemistry and World History was also conducted. The findings showed a statistically significant gain in student academic performance with respect to Chemistry and World History content. Students and teachers responded with overwhelming positivity and engagement to the intervention presented in this study. Future studies will focus on a more explicit integration of Engineering competencies and alternative assessments of academic learning outcomes. More importantly, the gap identified in this pilot study was the lack of time dedicated in the curriculum for students to learn about local cultural and to apply their new knowledge gained from CReST to undertake real-world and local problems defined as the summit of convergence education. However, as a whole, this pilot presents significant steps toward connecting STEM with culturally relevant pedagogy as a convergence education framework focused on societal challenges.

## 2. Literature Review and Theoretical Framework

### 2.1. Transdisciplinary and Convergence Education in STEM

One of the key pathways to provide a more meaningful STEM experience focused on real-world problems is "engaging students where disciplines converge" [12]. Although STEM education fundamentally involves multiple disciplines, the effective approach to the challenge of curricular integration is still unclear. The levels of integration can be differentiated into disciplinary, multidisciplinary, interdisciplinary, and transdisciplinary. The highest level of integration is transdisciplinary learning where students apply knowl-

edge from two or more disciplines through real-world examples [13,14]. Transdisciplinary learning can also be framed as convergence education. According to the U.S. Interagency Working Group on Convergence, "convergence education is driven by compelling or complex socio-scientific problems or topics, where learners apply knowledge and skills using a blended approach across multiple disciplines (i.e., transdisciplinary) to create and innovate new solutions" [8] (p. 7).

To investigate the effects of transdisciplinary learning in traditional settings, findings have shown different combinations of disciplines such as social studies and literacy with science and mathematics and physics with biology, which do not have a negative impact on the core subjects within the classroom [15,16]. Strategies to increase student self-efficacy in STEM subjects and career motivation have been pursued through the integration with Arts [16]. To address global issues within classrooms, work has been carried out to shift the learner experience from a passive to an active role. This work agrees that a transdisciplinary learning approach engages learners while simultaneously producing workforce readiness skills [17,18].

The complexity of the comprehensive approach to convergence education in terms of assessment is still unclear. Comprehensive reviews of interdisciplinary STEM education studies have been conducted to identify the types of assessment instruments [19,20]. Future research is needed in the development of assessments for interdisciplinary STEM education as it moves beyond the traditional educational model and the assessments mandated by educational systems at the state and local levels [8].

### 2.2. Experiential Learning in STEM

Kolb [21] describes Experiential Learning Theory as grounds for the learner to take knowledge and transform it into an experience within a wide range of situations. The combination of experiential learning with STEM education can assist with helping students not only better understand, but also apply their knowledge to real-world activities. Lestari [22] conducted a quantitative study on increasing problem-solving ability with physics high school students through experiential learning using a STEM approach. The results from a pre/post design showed a high level of student increase in problem-solving ability following the intervention. Long, Yen, and Hanh [23] explored the positive role, from the student perspective, of integrating the engineering design process with the Kolb model in a middle school STEM education setting.

Studies implementing modular STEM learning as a method of enrichment of the traditional curriculum [24,25] have been shown to increase student learning. These studies show a positive correlation to implementing STEM modular lessons within science classes to incorporate more real-world application in the learning experience. Yet, there is still a need to identify both qualitative and quantitative metrics for teacher knowledge, skills, and assessment in order to facilitate STEM modular learning in agreement with the set standards and pace.

### 2.3. Culturally Relevant Pedagogy in STEM

The purpose of culturally relevant pedagogy is to empower students [26]. Ladson-Billings [27] proposed that culturally relevant pedagogy should have three main components: academic success and student learning as the focus; the development of cultural competence to assist students' social identities while also gaining knowledge of other cultures; and skills to identify and solve real-world problems that have resulted in societal inequalities. This theoretical model assists students in acknowledging not only their own cultural identity but also that of others. Young et al. define culturally relevant STEM as "the utilization of cultural funds of knowledge inherent in all learners to develop deep and meaningful connections between STEM content and the learners' lived experiences" [28].

A multiple-case study was conducted to explore how the implementation of project-based learning through culturally relevant pedagogy in a science class would increase student learning and engagement known as invention-based learning (IBL). The results

showed that each participant student required a different amount of content knowledge prior to developing the cognitive ability to transfer that knowledge to real-world situations through a cultural lens. The findings illustrated that knowledge transfer is an "iterative bi-directional process", not one-way [29].

The Science Genius intervention implemented a qualitative study into ten urban high schools over an academic semester. The schools that participated in this study had students who underperformed, with a lack of interest in science. Science Genius is designed to target disengaged youth through a hip-hop-themed science program. The analysis produced three main themes: (1) the intervention revealed student emotions, (2) students' raps displayed evidence of essential science content knowledge, and (3) students' self-science identity was reframed through the intervention. This intervention allowed students to reveal their own thoughts and emotions while simultaneously reframing their own self-identity in the field of science. This research also displayed student evidence of essential science content knowledge [30].

Brown et al. [31] report that culturally relevant teaching remains stationary within STEM education. Some exceptions to this include the workings with hip-hop pedagogy and Science Genius [30,32]. The studies mentioned above showcase how equity pedagogies have been applied into STEM education. Collectively, they utilized qualitative studies with smaller populations to investigate their interventions. Though the overall findings suggest the positive effect of integrating culturally focused pedagogies into STEM, there is still a lack of quantitative research on how this impacts student learning.

*2.4. Intersecting the Four Axes through the Theoretical Framework of CReST*

As with the studies discussed above, there is an overall gap in research in terms of quantitative studies. The ability to gather quantitative data concerning student academic learning in alignment with the curriculum within any, or all, of the four areas of STEM would provide better evidence of the impact of the various pedagogical approaches in convergence education and culturally relevant pedagogy in STEM education. As the literature discussed, a common challenge within STEM education is the integration of multiple disciplines. Following the integration of STEM comes the assessment and the challenges of overcoming the traditional model of evaluating student academic success [8]. Lastly, student interest in STEM is of importance and there is an urgency to better meet the workforce demands across the globe. In response to the intersection of these gaps within STEM education, we investigated current interactive STEM frameworks as a starting point to develop a theoretical framework that provided a format for convergence education. Leung proposed a pedagogical framework that encompasses connective factors that produce boundary-crossing STEM pedagogy [33]. Leung's formulation of the Interactive Framework for STEM Pedagogy was built from research and findings on the challenges and obstacles an educator runs into as they attempt to cross from one domain in STEM into another. Situated learning, communities of practice, problem-solving, and learning dialogical processes are the four factors of Leung's pedagogy model. Situated learning concerns the cognitive processes of the teacher and student to understand how skills and knowledge can be applied.

Given the nature of the CReST approach to teaching and learning, we adapted Leung's Interactive Framework for STEM Pedagogy as a model within both the curriculum development and the assessment of the support curriculum [30]. The framework for CReST leverages cultural heritage artifacts, specifically fresco-style art, as boundary-crossing objects that thread together STEM disciplines through the usage of four factors, namely, Convergence Education, Communities of Practice, Culturally Relevant Pedagogy, and Experiential Learning, as shown in Figure 1.

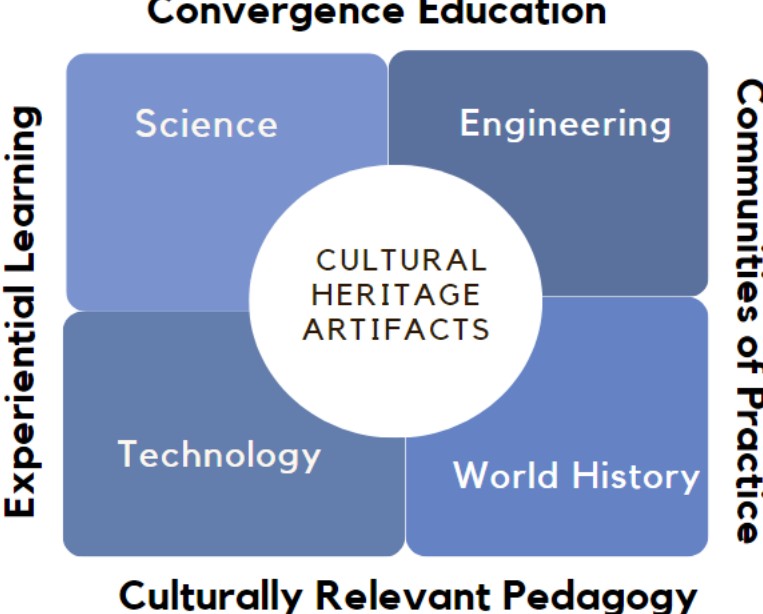

**Figure 1.** Adapted and modified version of Leung's Interactive Framework for STEM Pedagogy as applied to the CReST curriculum developed by the CReST research team.

The middle terms were modified from the original Leung figure from "boundary object" and "boundary-crossing" to "artifacts of cultural heritage" which is the object used within the CReST curriculum as the boundary-crossing object. The factor Situated Learning was adjusted to Convergence Education, and the other factor, Learning Dialogical Processes, was altered to Culturally Relevant Pedagogy to better align with the CReST curriculum and mission. Problem Solving Processes were adjusted to Experiential Learning. The teachers' community of practice was applied to CReST as the four participants worked together to implement their experiences from professional development.

The artifact of cultural heritage (i.e., frescos) serves as the common theme across the various disciplines to allow time and space for the transfer of knowledge to occur within the classroom. The CReST study used Leung's Interactive Framework for STEM Pedagogy (2020) concept of a boundary-crossing object to serve as the intersection.

In our pilot, we also leveraged the concept of convergence education and the focused problem space of cultural heritage conservation and the transdisciplinary intersections of science, engineering, and world history. We framed the didactic approach as culturally relevant pedagogy (CRP) because the integration of CRP is part of the CReST curriculum and mission. We highlighted experiential learning as a key feature of our approach to showcase how the CReST approach provides hands-on experience addressing real-world problems in a student-centered mode. We leveraged connections with communities of practice as an important aspect of teacher professional development through engagement with regional and international curators, clinicians, and research scientists at the Chrysler Museum in Norfolk, Virginia, the Center for Colloids and Nanoscience (CSGI) at the University of Florence, Italy, and Virginia Commonwealth University. As discussed by Herr et al. [34], relationship development with communities of practice is a key element of K-12 convergence education.

## 3. Materials and Methods

### 3.1. Research Design

To better capture the experiences of the CReST learning approach that leverages artifacts of cultural heritage, a mixed methods approach was selected to triangulate the data from assessment results with student focus groups and teacher interviews [35]. A quasi-experimental research design was used to evaluate the intervention without the

usage of randomization. The appropriateness of this type of research was for studying settings, such as already defined classrooms with an established teacher, prior to the intervention [36].

The research questions that led this study are as follows: 1. What were the experiences from the teacher and students with the integration of conservation science into their classrooms; 2. To what extent, if any, does the integration of conservation science with the Italian Renaissance in Chemistry and World History courses increase student academic performance in Chemistry and World History? For research question 2, the hypothesis that guides this question is as follows: $\mathbf{H_{a2}}$. $\mu_d \neq 0$ Students will demonstrate a greater learning gain in Chemistry and World History content knowledge as measured by state standards of learning instruments through the completion of a support curriculum that integrates conservation science.

To answer the first research question, a combination of a priori and inductive coding was applied to reach themes throughout the data [37]. To answer the second research question, statistical analysis was used to test the hypothesis.

### 3.2. Participants

We invited approximately 475 students from a mid-Atlantic US high school who were currently enrolled in Chemistry and AP Chemistry, and students from one World History I course to participate in this study. The demographic make-up of the student participants included males (53%), females (44%), White (35%), Black (26%), Asian (20.2%), and Other (17%). There were 276 Chemistry participants, generally sophomores and juniors, and 19 World History I participants. The World History participants consisted of all freshman students that had not yet taken a Chemistry course.

The instructional participants in the research study included three Chemistry teachers and one Social Studies teacher. Informed consent was obtained from all subjects involved in the study. The study received Internal Review Board approval prior to implementation.

### 3.3. Learning Intervention

3.3.1. Instructional Support Materials and Teacher Professional Development

We developed the CReST curriculum using a convergence education approach based on cultural heritage conservation and the (physicochemical) life cycle of mural paintings in Western Europe across 600 years of history. Alignment with state learning objectives in World History and Chemistry was considered during the development of CReST (Table 1). These learning objectives were selected based on the focused content of CReST and where the convergence of these two disciplines naturally aligns.

**Table 1.** State learning objectives in Chemistry and World History in CReST curriculum.

| Chemistry Learning Objective | | World History Learning Objective | |
|---|---|---|---|
| CH.7 The student will investigate and understand that thermodynamics explains the relationship between matter and energy. | | WHI.15 The student will apply social science skills to understand the developments leading to the Renaissance in Europe in terms of its impact on Western civilization by | |
| (a) | heat energy affects matter and interactions of matter; | (a) | determining the economic and cultural foundations of the Italian Renaissance; |
| (b) | heating curves provide information about a substance; | (b) | sequencing events related to the rise of Italian city-states and their political development, including Machiavelli's theory of governing as described in *The Prince*; |
| (c) | reactions are endothermic or exothermic. | (c) | citing the contributions of artists and philosophers of the Renaissance, as contrasted with the medieval period, including Leonardo da Vinci, Michelangelo, and Petrarch. |

**(a)**

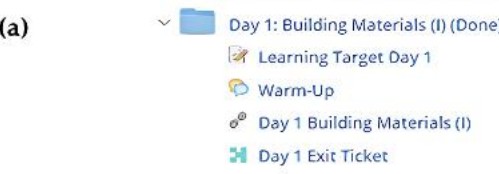

**(b)**

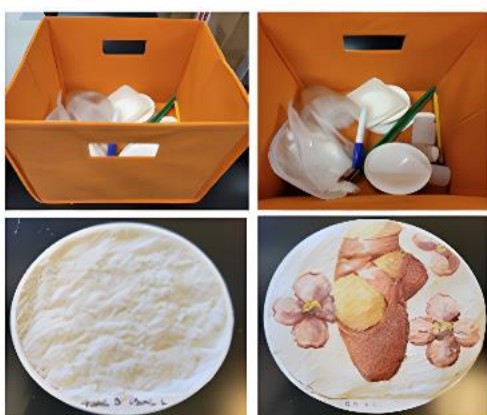

**Figure 2.** Instructional materials for CReST support curriculum: (**a**) plug and play learning modules and teacher support materials using the Schoology platform; (**b**) experiential learning kits with consumable supplies and tools to support individual student experience in fresco conceptualization, preparation, and conservation.

Each module lesson comprised of a daily learning target, Google Slide Decks for direct instruction of the content, and experiential materials along with warm-up and exit assessments that integrated state standards of learning (Figure 2). The development of the six instructional days of CReST originated from the life-cycle creation of the artifact used in this study, the fresco. The topics selected for each of the days directly align with the experiential activity that is partnered with the content (World History, Chemistry,

and Engineering), presented in each of the daily Google Slide Decks as the convergent instructional time. To create a fresco and then go through the conservation process, a minimum of six days of learning was required for schools that meet every other day for an eighty-five-minute class period. The experiential learning kits included 4″ circular tile molds, Rubbermaid dish pans, 50 mL disposable beakers, soft graphite pencils, gloves, small plastic cups, paint brushes (small and large), rice paper, Ziplock bags of Arbocel [38], disposable pipettes, pasty dishes, and Isopropyl Alcohol. Classrooms were also provided with washed sand, hydrated lime putty, and pigment. The curriculum and resources were housed in the participating school division learning management system, Schoology.

Teacher professional development for the participants was delivered through various modes of instruction and support. Four months prior to the intervention, the teachers participated in a Zoom meeting with the research team to discuss the overarching objectives and goals of the curriculum and intervention. This was followed by a full-day workshop at a local museum six weeks prior to the intervention, where they were provided with a summary introduction to each of the six instructional days and hands-on experiences associated with CReST. Virtual and in-class support sessions were offered to the teachers following the workshop as well as during the classroom implementation phase.

### 3.3.2. High School Classroom Implementation

The CReST support curriculum was executed in the participating Chemistry and World History classrooms for six instructional days, approximately eighty minutes per day, during the last two months of the academic cycle. Each day of the curriculum intervention included a *STEM Context*, *Experiential Activity*, and *Historical Context: Great Thinkers* (Table 2). The overarching learning target across the six-instructional-day intervention was as follows: Students will study the materials, design processes, environmental degradation, and conservation of Italian Renaissance fresco paintings to connect the concepts of Chemistry and World History with the application of Engineering and understand the conservation of our own cultural heritage.

**Table 2.** Overview of the six instructional days in the CReST curriculum.

|  | Day 1 | Day 2 | Day 3 | Day 4 | Day 5 | Day 6 |
|---|---|---|---|---|---|---|
| STEM Context: | Chemistry: Energy | Design | Chemistry: Reactions | Chemistry: Solutions and Dispersions | Environmental Chemistry | Nano chemistry |
| Experiential Activity: | Building Materials (I) | Sinopia | Building Materials (II) | Fresco Painting | Fresco Degradation | Fresco Conservation |
| Historical Context: Great Thinkers | Brunelleschi | Leonardo | Vasari | Michaelangelo | Ferroni | Baglioni |

This curriculum included Engineering, the bridge between Math and Science and the Consumer. This intervention enables the students and teachers to connect the fundamental element of inorganic chemistry, reaction equilibrium and kinetics, and nano chemistry to society issues such as challenges in cultural heritage conservation. Convergence education leverages an interdisciplinary approach to explicate the problem and develop solutions to societal challenges. CReST is a case study of convergence research in action and provides a local mechanism through engage with the communities of practice (higher education institutions, and local and state museums) to engage the students in personally relevant societal needs.

To further align with state learning objectives and instructional classroom norms of the participating school division, learning targets were established for each of the days. A brief description of each of the six instructional days of CReST is provided below.

Day 1 of CReST provided the students with an overview of the dark age and the rebirth of the Italian Renaissance. Day 1 introduced the dome of the Cattedrale di Santa Maria del Fiore constructed by Filippo Brunelleschi in Florence, Italy. One of the most remarkable structures of the Renaissance, completed in 1436, it features the largest masonry dome standing today [39]. Students were also introduced to fresco-style painting and the mural "Last Judgement". Frescos are painted onto freshly prepared lime plaster, usually within the same day.

Frescos are prepared on multilayer cement structures composed of calcium carbonate ($CaCO_3$—limestone) and silicon dioxide ($SiO_2$—sand). See Figure 3.

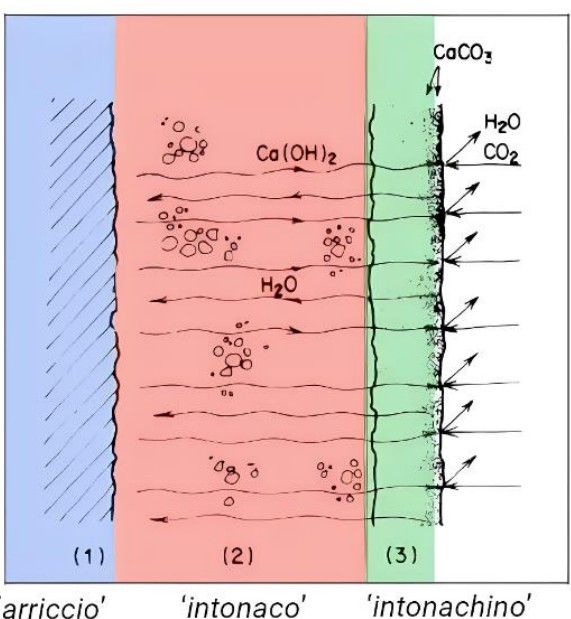

**Figure 3.** Anatomy of a fresco:Layers (1) arricio, (2) intonaco, and (3) intonachino. The multilayer (1)–(3) underlayment of a fresco is essentially a gradient of grasselo ($CaOH_2$—slaked lime) and sand. Arricio is approximately 1:3 (grasello/sand) and intonachino 1:1. Over time, the grasselo reacts with ambient carbon dioxide to form a lime crust ($CaCO_3$).

The students explored the fresco-making process through the chemical composition of the building material. To prepare fresco underlayment, calcium oxide (CaO), or 'calce viva', is manufactured through the thermal decomposition of natural materials such as limestone or seashells, which contain calcium carbonate [40]. With cement as the foundational material of the cultural heritage artifact, the fresco, Day 1 integrated relevance through discussion and calculations of the endothermic cement manufacturing process with the annual world energy consumption. Experientially, the students prepared the first layer of their fresco underlayment, arriccio, by mixing grasello (hydrated lime putty) and sand (Figure 4).

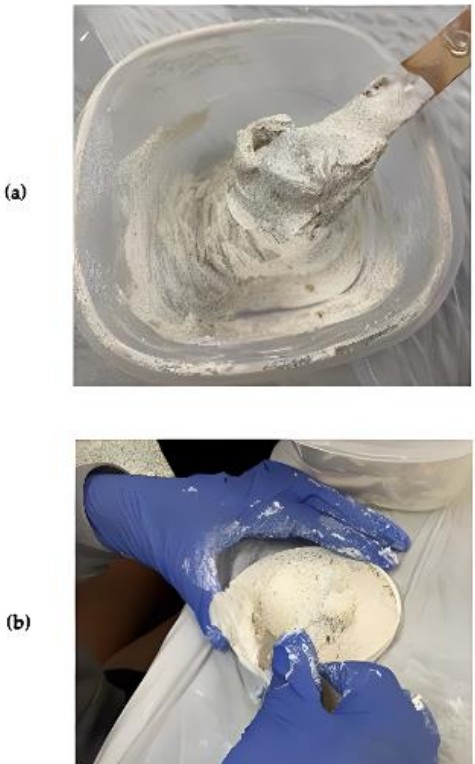

**Figure 4.** Day 1 experiential learning activity: (**a**) preparation of the mixture of grassello and sand; (**b**) preparation of arriccio using a disposable mold.

In Day 2, the students explored the underdrawings of a fresco, known as sinopia, which is made on the first layer of plaster (arricio/intonaco). Sinopias provide rare insight into the design thinking of the Renaissance masters; see Figure 5 for an example. We highlighted the panel painting "Adoration of the Magi" and Leonardo as the great thinker in Day 2 to help the students understand the progression from ideation to finished work. Day 2 took the students through the discovery of sinopia and its chemistry ($FeO_3$—sinoper/ferric oxide) [40]. The students participated in a self-reflection of their own culture and were encouraged to sketch a meaningful representation of their own interests or heritage onto their individual fresco plaster, as shown in Figure 5c.

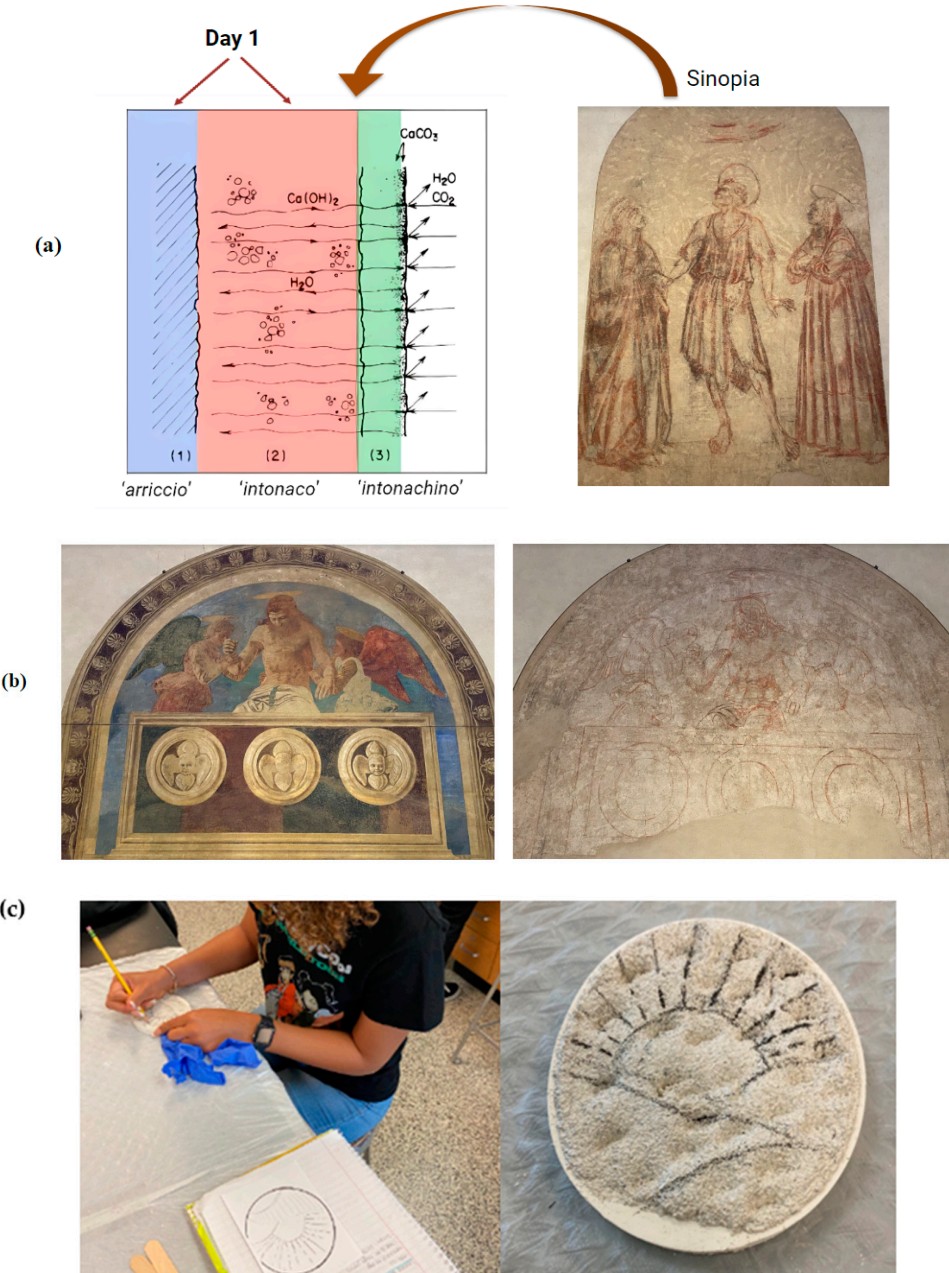

**Figure 5.** Day 2 design. Sinopia are the underdrawings of a fresco: (**a**) lecture component of Day 2: discussion of the design process with examples of sinopia and fresco; (**b**) sinopia at the convent of Sant' Apollonia, Firenze, Italy, was discovered during the conservation of the fresco; (**c**) students designing their own sinopia in the experiential component of Day 2.

Day 3 exposed the students to an early account of art history through *The Lives of the Painters, Sculptors and Architects*, by Giorgio Vasari [39]. The physical chemistry of the hydration of calcium oxide and the kinetics of the formation of the lime crust were explored through the carbonation reaction. Preparing slaked lime is a lengthy process because hydration is slow. Although the stoichiometry for preparing grassello (or lime putty) is 1:1 on a molar basis, there is about 30–40% free water that was traditionally prepared by the artists through a slaking reaction using a slight excess of water compared to the stoichiometric ratio. Lime putty contains finer particles with a high surface area. This gives rise to a higher reactivity and plasticity of the grassello with respect to the dry hydrate. The students made calculations regarding the recipe using the molecular weight, density, and

volumetric proportions. Experientially, the students applied the knowledge of the chemical reactions with the building materials to prepare the fresco for painting. They mixed sand with grassello and spread a thin layer (3–4 mm) of this malta (mortar) to obtain the pictorial layer or 'intonachino' which was spread on top of the sinopia from Day 2 (Figure 6).

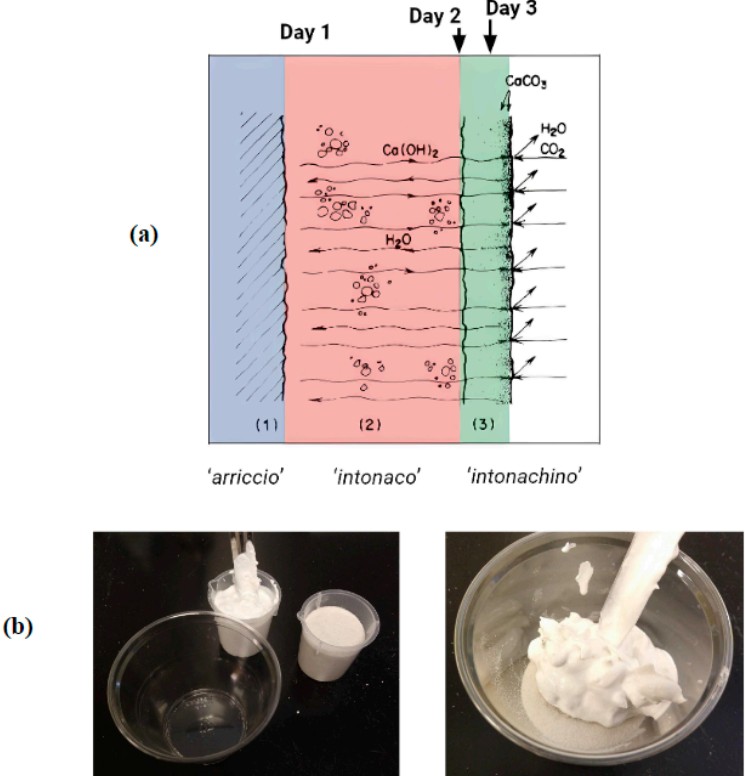

**Figure 6.** Day 3 curriculum of the intonachino layer: (**a**) illustration of the fresco layers; (**b**) mixing of the grassello and sand.

Day 4 featured Michelangelo as the great thinker with a discussion on the significance of the Basilica di Santa Croce in Florence, where his tomb and those of others such as Galileo Galilei and Niccolo Machiavelli are located [41]. The lesson differentiated a dye from a pigment through the science of a solution and dispersion and how that applied to fresco painting. Dispersions are two-phase systems consisting of a microscopic, dispersed phase distributed in a continuous, matrix phase. Because dispersions are thermodynamically unstable, they require an addition of energy to maintain the dispersed state. The destabilization of a dispersion happens mainly because gravity acts on the dispersed phase and causes it to sediment (or cream) and separate. The students then painted their frescos by applying an aqueous dispersion of inorganic pigments (Figure 7).

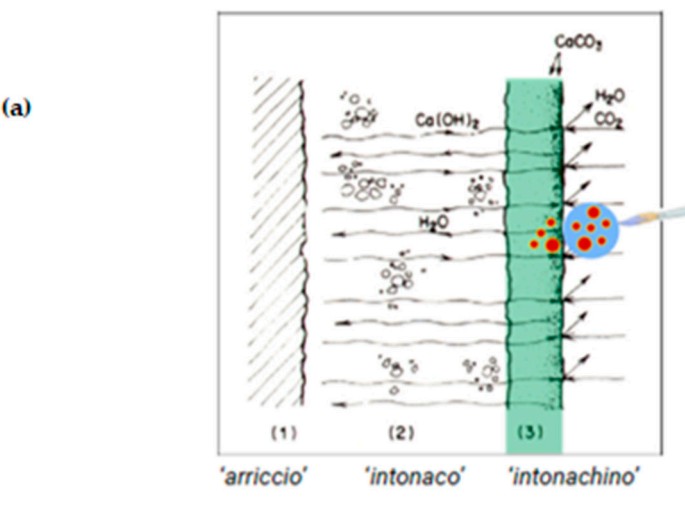

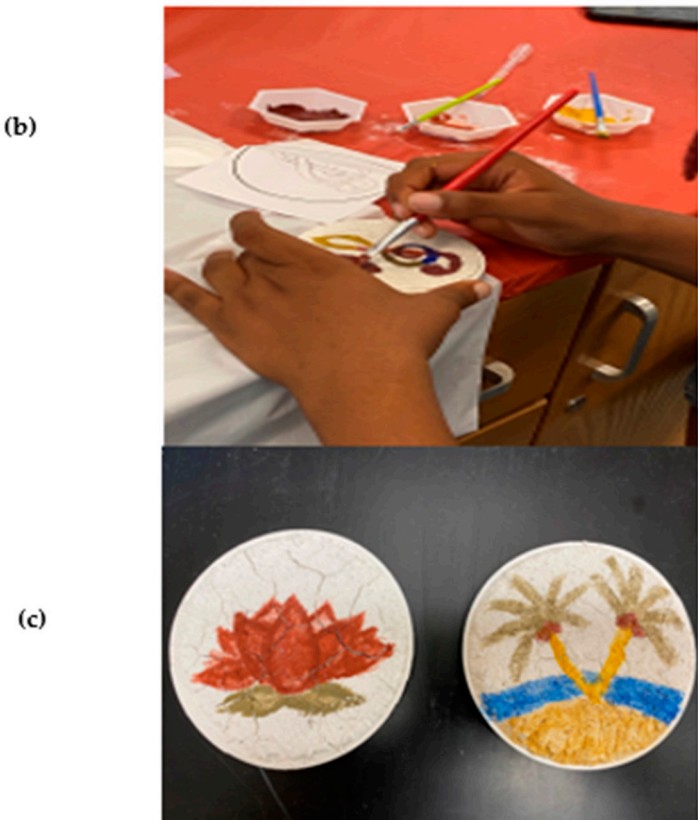

**Figure 7.** Day 4 experiential learning activities of the CReST curriculum: (**a**) schematic of the pigments suspended in acqua di calce (lime water) as applied with a paint brush; (**b**) a student painting their designed fresco with pigment; (**c**) two examples of student fresco paintings.

The last two days of the CReST curriculum discussed cultural heritage conservation. Day 5 focused on environmental chemistry. The Flood of Florence in 1966 provides the historical context. During the Flood, water levels reached 22 feet in Santa Croce, damaging more than 500,000 cultural heritage artifacts in a single event [42]. To understand some of the anthropogenic sources of fresco degradation, students were introduced to the chemistry of fossil fuels (natural gas, oil, and coal). Crude oil that contains very little sulfur is called sweet; Brent sweet crude oil is sweet. Crude oil that contains sulfur is called sour. When these fossil fuels are used as energy sources, they are oxidized (burned). When the sulfur

is oxidized, it forms sulfur dioxide ($SO_2$). This can react with the fresco ($CaCO_3$) to form selenite ($CaSO_4$). The formation of selenite can damage the fresco. Prior to the Flood of Florence, frescos were conserved using detachment (the strappo technique) and ex situ laboratory treatment [43,44] to mitigate the deleterious effects of selenite formation of the mural surface.

The featured great thinker in Day 5, Enzo Ferroni, is the co-discoverer of the famous Ferroni–Dini method, which was the first in situ technology for conserving a fresco. Ferroni was one of the first scientists to apply a scientific approach to the conservation of cultural heritage to help save Italian masterpieces. He is also the founder of the Consorzio Interuniversitario per lo sviluppo dei Sistemi a Grande Interfase (CSGI)—an academic community in Italy [45].

The students learned about the effects of anthropogenic (man-made) pollution and the chemistry behind the degradation and conservation of fresco art.

Day 6 focused on contemporary conservation approaches and the use of nanotechnology for the conservation of frescos. Piero Baglioni, a former student of Enzo Ferroni, and current President of the CSGI was highlighted as the great thinker for Day 6 [46]. The students learned about the properties and application of calcium hydroxide ($CaOH_2$) nanoparticles to conserve a fresco [43,44]. Experientially, the students practiced the process of conserving their fresco (Figure 8) using a classical conservation approach.

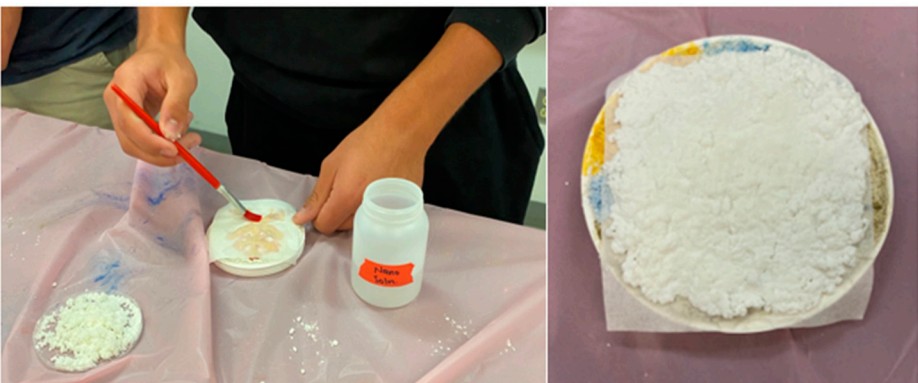

**Figure 8.** Day 6: the students apply $Ca(OH)_2$ through Japanese rice paper and damp Arbocel onto their fresco to complete the conservation process.

### 3.4. Data Collection and Analysis

The data sources for this pilot study included pre- and post-assessments, student focus groups, and semi-structured teacher interviews. The data analysis employed both qualitative and quantitative approaches. The teachers had completed consent forms prior to the intervention. An approved letter from the school division was sent home with the students prior to the implementation of the six instructional days of CReST where they were able to opt out of the research components of the intervention.

#### 3.4.1. Qualitative Data

To capture the perceptions of the CReST intervention, student focus groups and semi-structured teacher interviews were conducted approximately two to three weeks following the conclusion of the six instructional days of CReST. The selection of the student participants in the focus group followed a process to ensure diverse student voices by representation from the three different chemistry teachers as well as gender and ethnicity diversity. The students were assembled into two different focus groups that were led outside of the classroom and the meetings were conducted in a conference room. To gather the experience of the students, the focus groups used semi-structured open constructed responses with the usage of an interview guide. The constructed responses included six questions surrounding their thoughts on the six instructional days of cultural heritage

conservation and how that affected their learning and interest in STEM. The facilitator then asked, one at a time, each of the questions (see Appendix A).

All four teachers that attended the professional development workshop and implemented the six instructional days participated in semi-structured interviews with open constructed responses [47] via Zoom; see Appendix B.

The student focus group and teacher interview data were transcribed and analyzed using inductive coding. Two rounds of coding (initial and focused) occurred to identify trends and overlaps [48] from the triangulation and comparison of the data given to the various groups of participants.

### 3.4.2. Quantitative Data

To assess the potential of learning gain, the students were administered an identical pre- and post-assessment composed of state-released questions from Chemistry and World History I and II. Ten of the questions focused on Chemistry and six on World History. The assessments were administered within ten days prior to and following the implementation of the CReST intervention. The students were given the first 20–25 min of class to complete the pre- and the post-assessment.

To analyze the quantitative data of the pre- and post-assessment, the responses were paired only with students that opted to be a part of the research study and completed both the pre- and post-assessment. The data were aggregated to separate the Chemistry student responses from the World History I students. The aggregate score on the pre- and post-intervention instruments with stratification by each subject matter (e.g., Chemistry, World History, Engineering) subset was then analyzed through a paired *t*-test with IBM SPSS Statistics Base version 28.01 [49]. Time, the independent variable, was used as the within-subjects factor to measure any main effects. The dependent factors were the performance scores from the pre- and post-assessment.

### 3.5. Limitations

The results of this study can be interpreted through several limitations. First, this intervention only lasted six instructional days inside three different high school Chemistry and one World History I classrooms. All the courses were year-long in duration. If the treatment lasted longer and was integrated into more units, the treatment effects observed may be larger on Chemistry and World History.

In terms of fidelity, across fifteen different Chemistry classrooms there were three different teachers that implemented the CReST curriculum. While they were provided with the same professional development and instructional resources, each teacher's instructional delivery varied based on the individual students' needs and their own pedagogical style. Though instructional support was provided, there was some variation across the teachers and sections with the pacing and focus with each of the lessons.

The variability of student attendance is another limitation. If a student was absent on any of the six instructional days of the intervention, it could affect their post-assessment content score.

The participants in this study were limited to one high school in a suburban school division in a mid-Atlantic state. In the participating school division, not all students are required to take Chemistry as a graduation requirement. This course is considered a higher level course and tends to enroll higher performing students. With that said, all students are required to complete World History I as a graduation requirement. Given the timing of the intervention with state standardized testing, the World History I students were not included in the student focus groups.

### 3.6. Validity and Reliability

To avoid biased interpretations and alleviate researcher bias of the qualitative data, member checking was implemented for the teacher interviews and student focus groups to verify the themes from the research analysis.

For the assessment, the pre- and post-assessments contained state-released exam questions in both Chemistry and World History I and II. Both the pre- and post-assessments contained identical questions.

## 4. Results

### 4.1. Qualitative Findings

The first research question probed at the experience of the teachers and students in relation to the CReST curriculum intervention. Our qualitative analysis separately analyzed the teacher's perspectives and the students' perspectives, and then converged the overall experiences. There were four major themes that emerged from both populations: (1) Hands-On, Learn by Doing; (2) Real-World Experience, Application, Relevance; (3) Cross-Curricula, Interdisciplinary, Connecting Concepts; and (4) Global/Culture. We discuss each of the four themes below.

### 4.1.1. Teacher Experiences

Two of the teacher interview questions inquired about the effectiveness of the workshop and usefulness of the curriculum support activities. Each of the teachers' responses included the effectiveness of the *hands-on* element regarding both the teacher PD workshop and learning activities within their classrooms. In response to which components were effective in the classroom, one of the teachers stated, "*The most effective ones were all the hands on, that was huge. Because the kids were able to apply what they were learning and then do what they were learning*".

When asked about the correlation between the transdisciplinary approach and the teachers' perspective on student learning, the common theme was *connecting concepts*. They shared that the daily great thinkers integrated the science and history concepts. The teachers all mentioned implementing a daily recap of the previous lesson(s) as a strategy to bind together the three overlapping disciplines. They shared that it was extremely helpful with the interdisciplinary/transdisciplinary model, which was not discussed in the professional development workshop. One of the chemistry teachers shared, "*Before we would start every lesson, we did a little brain dump of everything we learned the previous day. So, in the lesson itself, because there's so much content, I don't know how much they grasped, but then we all brain dumped the following class and connecting it each day and bringing it back to the main theme*".

Another theme that emerged from the transdisciplinary experience was the connection to culture and global learning experience. A chemistry teacher explained, "It was good for obviously the students, but it was also good for us teachers. I feel, at least for me, I can't speak for the other ones, but I know when we've spoken, they said that they enjoyed digging in deeper, looking into more information about, you know making it global".

The teachers were also asked how the curriculum could be altered to better support student learning goals. Some of the suggestions included the addition of a vocabulary word bank (given many of the words were in Italian), more opportunities for student discussions, and increasing the image sizes of famous Renaissance artifact works on the content slide decks. A consistent theme across the teacher interviews was their interest in including other cultures and artifacts from various parts across the globe. They expressed that the format of the CReST curriculum could lend itself to the student exploration of more cultures.

The final question inquired whether the teachers would apply the CReST curriculum into their classroom again. The responses resurfaced all the major themes. One of the teachers stated, "*Oh, it was so fun. I loved it 100 percent. I think they got more out of it seeing that everything isn't this individual idea, but in real world, everything is interconnected*". The other three teachers had the same enthusiastic response to implementing the CReST curriculum again within their classes. The World History teacher shared, "*I saw students who probably never thought about taking chemistry before, excited about it. I had other students that thought they were going to struggle say, 'I get this'*".

4.1.2. Student Experiences

During the student focus groups, when asked to share what they liked about the CReST intervention, they all provided examples of the *hands-on* components of the curriculum. This was consistent with the teachers' responses. One student expressed their enjoyment with Day 3 Painting by sharing, "*I really enjoyed doing the painting itself because once we got the pigment to lime water ratio right, it was really easy to do. And mine came out really pretty*". Another student shared how much they enjoyed the entire experience and explained how other classes were talking about it by stating, "*Yeah. I know other classes who didn't get to do the frescos were like, 'Oh man, I wanted to do that'*".

When asked about which parts of the CReST curriculum helped their learning, the common theme among the students were the content slide decks at the beginning of each lesson. The students explained that they *blended* the content together and created *relevance* in their learning environment. One student shared, "*I think learning about the history portion and seeing what other people did and modifying it to make it so we can do it in chemistry*".

Students were then asked whether they would like other lessons to be taught with the overarching theme of cultural heritage conservation to integrate STEM. All students agreed that they enjoyed the structure and that it allowed them to understand the *application* of chemistry by learning another *culture.* One of the students shared, "*Yes. This was so much fun and I really enjoyed it and I feel like I really learned the concepts thoroughly. Other cultures that I would like to see would probably be more of the Middle East and African cultures. Because I feel like those aren't really covered a lot in schools*". Another student commented, "*Yeah. I believe it is very informative approach and different because it's definitely a learning technique that most schools don't have the opportunity to take*".

Finally, the students were then asked how the STEM (CReST) curriculum support affected their interest and plans in pursuing a career in STEM. Seven out of the nine student focus group participants indicated that it had a positive impact on their interest and ability to pursue the STEM field. One student shared that their perspective of STEM was altered after the experience by stating, "*Oh, it's not just building robots and stuff. It's also going to include engineering things that are chemical, working hands on with stuff that's not this preconceived notion of what STEM must be*". A common theme throughout all the students' responses to the intervention was that is fun to learn about STEM in a different way that incorporated other disciplines and ideas.

*4.2. Quantitative Findings*

The second research question investigated whether the curricular intervention influenced student academic performance. The study separately analyzed the Chemistry and World History I student populations. The Chemistry sample showed a statistically significant learning gain in both Chemistry (4.0%) and World History I and II (13.4%) content ($n$ = 276, $p < 0.05$, paired $t$-test, d = 0.25), contradicting the null hypothesis for the Chemistry student population, as shown in Table 3.

**Table 3.** Descriptive statistics for the Chemistry student population on academic learning.

| $n$ = 276 | PRE-Test Mean | POST-Test Mean | Mean Difference | $p$-Value |
|---|---|---|---|---|
| Chemistry Questions (10) | 69.3 | 73.2 | 4.0 | <0.001 |
| World History I and II Questions (6) | 68.4 | 81.8 | 13.4 | <0.001 |

$p$-value set at 0.05.

The World History I student population data used the same statistical analysis approach (Table 4). This sample did not show a statistically significant learning gain in either Chemistry (1.6%) or World History I and II (10.5%) content. The $p$-value was greater than 0.05 which could be an effect of a small sample size.

**Table 4.** Descriptive statistics for the World History I student population on academic learning.

| *n* = 19 | PRE-TestMean | POST-TestMean | Mean Difference | *p*-Value |
|---|---|---|---|---|
| Chemistry Questions (10) | 32.1 | 33.7 | 1.6 | >0.05 |
| World History I and II Questions (6) | 55.3 | 65.8 | 10.5 | <0.001 |

*p*-value set at 0.05.

## 5. Discussion

The purpose of this pilot study was to develop, implement, and assess a new support curriculum, known as CReST, that leverages cultural heritage conservation in Chemistry and World History high school education on the fresco making process through convergence education. Current STEM educational research suggests the importance for students to solve real-world problems that are compelling and require the application of knowledge in a meaningful way [8–11]. Studies also suggest that implementing culturally relevant pedagogy in STEM classrooms can increase student interest in the field [28–32]. The results from this pilot study add to the literature through documenting a positive experience from both the teacher and student populations through a convergence STEM approach with the integration of culturally relevant pedagogy and experiential learning. A breakdown of the various components of this study through the theoretical framework is discussed below.

### 5.1. Alignment of Themes with the Theoretical Framework

Across both populations, the emergent themes from the qualitative analysis had a direct correlation with the four factors that bound our framework (shown in Figure 1). As summarized in Table 5, *convergence education* was positively discussed by both students and teachers as it created more relevance within the learning environment. As convergence education is ultimately defined through the highest level of integration, transdisciplinary learning [8], further support for teacher professional development on "how to" blend the delivery of the three contents of CReST was a result of this pilot study. The participant teachers found that a daily spiral review at the beginning of each class period assisted with bringing together content in a meaningful way. As this curriculum was taught in both Chemistry and World History classrooms, the teachers were able to support each other in their areas of expertise, except within the Engineering discipline.

*Experiential learning* was intentionally integrated into the writing of the curriculum. As the intention of Experiential Learning Theory is for the learner to transform their knowledge into a range of situations [21], the results showed that the teachers and students applied the content in Chemistry and World History through hands-on experiences in an unexpected yet relevant way to serve a greater purpose globally. The teachers overwhelmingly stated that the hands-on workshop was the most beneficial experience prior to implementing CReST into their classrooms. *Communities of practice* were also purposefully designed within the format and collaboration of this project. The teachers benefited from working closely with a chemical engineering professor to increase the depth of their knowledge surrounding the subject area. The teacher PD workshop at the local museum showcased how disciplines can be unified. Also, the teachers themselves worked together as a community of support. As stated, the development of communities of practice is suggested to support convergence education as it enables educators to share their best practices and challenges [34]. *Culturally relevant pedagogy* allowed for the exploration of a different culture and how that organically created classroom discussions about one's own culture. The data showed that the students and teachers were interested in applying the CReST approach to various cultures, not just the Italian culture. Student learning was assessed and achieved during this pilot study. Both teachers and students shared how they were able to identify the relevance within the curriculum as it applied to society. All three tenets of culturally relevant pedagogy [27] were identified throughout the intervention. A more in-depth investigation of each of the

three tenets as they apply to CReST would provide insights into how student and teacher thinking about culture shifted during this experience.

**Table 5.** Summary of the CReST curriculum findings through the adapted and modified version of Leung's Interactive Framework of STEM Pedagogy.

| Factors of Interactive STEM Pedagogy | CReST Curriculum in High School Chemistry and World History |
|---|---|
| Convergence Education | (1) Relevance to the learning experience occurred;<br>(2) Additional PD on "how to" best deliver three disciplines;<br>(3) Spiral recap of the previous lesson(s) to assist in blinding disciplines together;<br>(4) Need to further bring the Engineering discipline forward;<br>(5) Need to dedicate time for students to apply their new knowledge to undertake local problems as the final assessment of the CReST approach. |
| Experiential Learning | (1) Environmental factors of energy consumption with making cement;<br>(2) Conservation of artifacts;<br>(3) Application of nanotechnology. |
| Communities of Practice | (1) PD workshop at local museum met the conservators;<br>(2) Teachers benefited from working with a chemical engineering professor;<br>(3) Teachers worked together. |
| Culturally Relevant Pedagogy | (1) Chemistry students learned of a different culture unexpectedly in their science course;<br>(2) Teachers were exposed to incorporating cultural pedagogy into their learning environment;<br>(3) Students and teachers had suggestions on how to expand CReST to other cultures and artifacts;<br>(4) Student learning was achieved;<br>(5) Teachers and students alike identified and solved real-world problems. |

*5.2. Teacher Professional Development and Implementation*

Although the workshop was limited to a single day in this pilot study, all teachers expressed confidence in the ability to implement the lessons with their students. The teacher interviews described the effectiveness of ongoing support throughout the intervention as well as the hands-on activities during the workshop, which had extensively assisted in their level of confidence with facilitating the CReST curriculum within their classrooms.

An overarching theme of the Chemistry teachers was their instructional effort to incorporate Engineering and World History into their lessons. As for the Social Studies teacher, she had to dive deeper to make connections across curricula, especially with the chemistry content. The findings displayed a more immersive and extended duration of the PD would better prepare the teachers for the implementation of CReST into their classrooms.

*5.3. Student Engagement with the Topic and Achievement*

The student engagement within this intervention was overwhelmingly high as reported by all four teachers. The same level of enthusiasm was shared during the student focus groups. The students reported that they had never participated in a project that unified science, social studies, and engineering. The students used the word "fun" multiple times within the focus groups. World History I students expressed excitement about taking chemistry in the future, which speaks to the influence the CReST curriculum had on the students' view of science. Though a pre- and post-assessment of student STEM self-efficacy was not reported in this pilot study, all the student focus group participants vocalized an increased interest in pursuing STEM or a reinforcement of their aspirations to explore the field of STEM following the intervention.

To answer research question 2, the results showed an overall statistically significant learning gain from the Chemistry student population sample. The results indicated an increase in learning within both Chemistry and World History from both student participant groups.

*5.4. Assessment and Support Materials*

Given the comprehensive approach to convergence education, the terms of assessment are still to be identified [19,20]. During the initial phase of the CReST curriculum development, the alignment of the state standards of learning preceded the creation of the modular days. Preliminary data from this pilot study show that further curriculum development is needed to better integrate and align Chemistry state standards with the CReST curriculum. A more direct approach to teaching students the engineering design process should be applied to curriculum advancements. The pre- and post-assessment did not include questions surrounding the engineering design process. Future work could include an assessment of student learning regarding the engineering design process.

The validity of the administered pre- and post-assessment scores could be improved by using different pre- and post-assessment questions that align with the same learning objectives. Furthermore, the inclusion of an open-ended questionnaire would provide students and teachers with the opportunity to share their experiences during the teacher's professional development and over the six instructional days. This would provide data for the research team over the course of each instructional day, instead of only an overall experience.

The one major issue with the fresco making process was cracking. Following the Day 3 activity, intonachino, the frescos dried too quickly with the lack of humidity, which resulted in cracking. Nevertheless, the students were able to paint their frescos. Supplementary testing on the factors that affect the fresco drying process would need to be conducted prior to additional implementation.

The CReST curriculum required support materials for the experiential learning activities that are not typically found in a Chemistry or World History classroom. With that said, funding would be needed for the implementation of this support curriculum. Additional considerations include set-up, clean-up, and the storage of the materials. As mentioned by the World History teacher, Social Studies classes do not typically facilitate laboratory activities. This should be considered prior to the implementation of a Social Studies course also with regard to the space in which it is being conducted, such as a classroom with access to water and counter space.

## 6. Conclusions

Here, we describe an emerging convergence education approach through a CReST pilot program. The research questions focused qualitatively on the experiences of the teachers and students as well as quantitatively to assess whether there was an increase in student academic performance in Chemistry and World History. This intervention, a STEM support framework and curriculum, brought forward a methodology to learning that shifts away from mono-disciplinary, lecture-centric content mastery to create convergent learning

opportunities that are driven by compelling socio-scientific problems. Though the final stage of this pilot study did not prompt students to apply their new knowledge to solve real-world or local problems, an extension of the six instructional days could certainly allow for the teachers and students to make connections within their own communities.

Our pedagogical strategy was based on a blended, transdisciplinary, modular, experiential approach and provided a strong platform for convergence education. We found that the teachers and students were fully engaged in the six instructional days of CReST as it provided both stakeholders with a platform to integrate STEM and relevance within the learning environment.

We anticipate future work to be focused on enhanced teacher professional development to provide a richer cultural context and further connections with the communities of practice in the CReST curriculum. Future studies will facilitate teacher workshops for two or three days, with follow-up sessions both before and during the intervention, to best support teaching and learning. We foresee curriculum development and studies for global cultural heritage beyond the Italian culture, with artifacts from different geographical regions. To enhance the Engineering content, we plan to create more balance across the three blended disciplines.

In the next phase, we also anticipate that larger sample sizes will improve the resolution of learning gains from the implementation of the curriculum support in World History courses.

**Author Contributions:** Conceptualization, J.K.F. and R.S.W.; methodology, J.K.F. and R.S.W.; formal analysis, J.K.F. and R.S.W.; investigation, J.K.F. and R.S.W.; resources, J.K.F.; data curation, R.S.W.; writing—original draft preparation, R.S.W.; writing—review and editing, J.K.F. and R.S.W.; visualization, J.K.F. and R.S.W.; supervision, J.K.F.; project administration, R.S.W.; funding, J.K.F. All authors have read and agreed to the published version of the manuscript.

**Funding:** This research received no external funding.

**Institutional Review Board Statement:** The study was conducted in accordance with the Declaration of Helsinki and approved by the Institutional Review Board (or Ethics Committee) of Virginia Commonwealth University (protocol code HM20024331 and approval date May 4, 2022). "Enhancing the STEM Curriculum with Conservation Science: a case study of the Italian Renaissance and frescos" for studies involving humans.

**Informed Consent Statement:** Informed consent was obtained from all subjects involved in the study.

**Data Availability Statement:** Anonymized data are available upon reasonable request from the corresponding author.

**Acknowledgments:** The authors would like to thank Kaitlin Kay for supporting laboratory kit development, Emily Cayton for hosting the teacher PD workshop, the participating teachers and school division for their partnership and involvement with this project, and the reviewers for their feedback on this paper.

**Conflicts of Interest:** The authors declare no conflicts of interest.

## Appendix A

1. Which course are you currently enrolled in this year (Chemistry, AP Chemistry or World History I)?
2. What is the purpose of making the under-drawing (sinopia) if it ends up getting covered by intonachino?
3. What is the difference between paint and pigment?
4. Which activity in the STEM cultural heritage conservation modules did you like? Why?
5. While completing the STEM cultural heritage conservation activities (modules and hands-on lab activities), which parts of it helped you learn? Why?
6. Did you think that the STEM cultural heritage conservation activity modules had an effective connection and flow with chemistry and world history content?

7. How could the STEM cultural heritage conservation activity modules be altered to better support your learning goals?

**Appendix B**

1. Which course(s) are you teaching this academic year?
2. Which parts of the workshop did you find to be useful when implementing this learning approach in your classroom?
3. Which components of the STEM cultural heritage conservation modules and hands-on learning activities did you find to be effective in your classroom?
4. Which components of the STEM cultural heritage conservation modules and hands-on learning activities do you think assisted in student learning? Why?
5. Did you think that the STEM cultural heritage conservation activity modules had an effective connection and flow between chemistry and world history content?
6. How could the STEM cultural heritage conservation activity modules be altered to better support student learning goals?
7. Would you like to facilitate other lessons with the integration of transdisciplinary STEM cultural heritage conservation?

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
