# Peer review of "Culturally Relevant STEM (CReST): An Integrated Support Curriculum for High School Chemistry and World History"

_education, doi:10.3390/educsci14020182_

Round 1

Reviewer 1 Report

Comments and Suggestions for Authors

Revision “Culturally Relevant STEM (CReST): A Convergence Education Approach”

Top of Form

Comments and Suggestions for Authors

The title as well as the introduction raised expectations about your manuscript and research, but you should consider using the acronyms "(CReST)".

The topic you are addressing would be a relevant addition to existing literature, we need more research on this topic. Thank you for this contribution. I will structure my feedback in (a) general remarks (these comments cover feedback applicable in the entire manuscript), and (b) specific remarks (feedback on sentence and/or word level). The specific remarks can include a quote from your original manuscript to refer to a specific section. The specific remarks will refer to page (emphasis added in boldface; e.g., 1.15/16) and row(s; e.g., 11.15/16).

 General remarks:

The general manuscript addresses relevant topics of great importance to our society. Thanks for addressing this. However, some aspects need to be clarified and improved. Despite its potential, there are problems of coherence and concepts. As I read your work, I have many questions that remained unanswered in your manuscript. 

Specific remarks:

p.1                   The abstract does not mention the research questions or the objective. I propose a review of the key words, it seems to me that there are too many words and there are intersections.

p.2                   Shouldn't figure 1 be part of the theoretical framework? The adaptation made must be more in-depth and not left to the reader.

p.3-5               The  “Literature Review and Theoretical Framework” presents 4 axes: Transdisciplinary and Convergence Education in STEM; Experiential Learning in STEM; Culturally Relevant Pedagogy in STEM; Theoretical Framework of CReST. What is the integration between them in this work? Although some connections were established, I propose that this connection, the theoretical basis of the work, be improved.

p.5                   “To answer the research questions, two different approaches were applied. The qualitative components of the research questions used a combination of a priori and inductive coding to reach themes throughout the data [37]. To answer the qualitative components, statistical analysis was used to test the hypothesis through the statistical software of SPSS [38]. Is this what you want to say?

p.5                   “For reliability and validity, the pre-and post-assessment for the students contained state released exam questions in both Chemistry and World History I & II. Both the pre-and post-assessments contained identical questions. To avoid biased interpretations, member checking was implemented for the teacher interviews and student focus groups to alleviate researcher bias by allowing someone else to verify the themes from the qualitative research analysis.” The fact of having exam questions is what contributes to the reliability and validity of the study? It is not comprehensible...

p.7                   “The overarching learning target across the six-instructional day intervention stated: Students will study materials, design processes, environmental degradation, and  conservation of Italian Renaissance fresco paintings to connect the concepts of Chemistry, World History and Engineering”. What does the term “Engineering “mean? It is important to clarify, as it appears later (for instance “Engineering Design Process”)

p.13                 About 3.4.1 Qualitative Data, ““Given the timing of the intervention with state standardized testing, the World  History I students were not included in the student focus groups”, Is this a limitation of the study or not? It must be indicated as such.

p.17                 Table 5 should be revised. “Convergence Education “ 5)  without text.

The term convergent education is frequently used throughout the text. But I have doubts about whether we can say that there was a convergent education when the assessment is not at all aligned with the experience developed.

p.19 I propose that the research questions be revisited in the conclusions, even if in a synthetic way, to serve as a basis for the statements made.

p.References   there are some consistency issues in your reference list. I do not identify all references in the text (e.g. 49). You must check the coherence of the referencing.

Reviewer 2 Report

Comments and Suggestions for Authors

This is very interesting research and a job well done. For further improvement, these recommendations can be considered.

1. The abstract can specify in more detail the problem that motivated this study. The research objectives can also be added, while the section on methodology can be explained more briefly.

2. In the Introduction section, you can add:

·       The reasons for the choice and convergence of the two selected subjects.

·       A brief overview of the results obtained.

·       The goals formulated for future studies.

·       The gaps identified during this research.

3. In section 3.3.1 of the methodology, you can explain the reason behind the choice of learning objectives. In section 3.3.3, you can explain the reason behind the choice of conducting the lectures over six days and the selection of appropriate topics.

Author Response

Please see attachment for Author's Reply to the Review Report. Thank you.
